# The Undeniable Potential of Thermophiles in Industrial Processes

**DOI:** 10.3390/ijms25147685

**Published:** 2024-07-13

**Authors:** Giovanni Gallo, Paola Imbimbo, Martina Aulitto

**Affiliations:** 1Division of Microbiology, Faculty of Biology, Ludwig-Maximilians-Universität München, 82152 Martinsried, Germany; giovanni.gallo@lmu.de; 2Department of Chemical Sciences, University of Napoli Federico II, Complesso Universitario Monte Sant’Angelo, 80126 Napoli, Italy; 3Department of Biology, University of Napoli Federico II, Complesso Universitario Monte Sant’Angelo, 80126 Napoli, Italy

**Keywords:** thermophiles, anaerobic digestion, medium-chain fatty acids, biohydrogen production, biodegradable plastics, polyhydroxyalkanoates, high-added molecules

## Abstract

Extremophilic microorganisms play a key role in understanding how life on Earth originated and evolved over centuries. Their ability to thrive in harsh environments relies on a plethora of mechanisms developed to survive at extreme temperatures, pressures, salinity, and pH values. From a biotechnological point of view, thermophiles are considered a robust tool for synthetic biology as well as a reliable starting material for the development of sustainable bioprocesses. This review discusses the current progress in the biomanufacturing of high-added bioproducts from thermophilic microorganisms and their industrial applications.

## 1. Introduction

Thermophiles and hyperthermophiles are microorganisms able to survive in environments generally considered hostile to humans and other microorganisms, such as the deep sea, volcanic sites, and hot springs. The temperature range that these microorganisms can endure is from 45 °C to 75 °C for thermophiles and between 80 °C and 110 °C for hyperthermophiles [1]. Among all the environmental factors, temperature has provided insights into understanding biodiversity [2], the physiology of adaptation mechanisms [3], and consequently has allowed the reconstruction of the evolutionary history of life on Earth [4]. Compared to their mesophilic counterparts, thermophiles show complex genetic and physiological changes that evolved as a response to temperature-related stress. The protection mechanisms rely on several factors, such as the alteration in cell morphology, the activation of detoxification enzymes to counteract oxidative stress, the overexpression of heat shock proteins [5], and the presence of chemically stable lipids within the membranes, whose fatty acid composition is regulated by the increase in the external temperature, and, finally, the presence of thermostable enzymes/proteins within the microbial cells [4]. Structural studies revealed that amino acid substitutions found on the surface regions of thermophilic proteins are responsible for an increased number of intermolecular salt bridges and hydrogen bonds that provide the stabilization of the secondary and tertiary structures [6,7,8]. Over the years, the biotechnological potential of thermo- and hyperthermophilic microorganisms has been promoted as an alternative to the use of mesophilic ones in industrial processes. To be effective, biotechnological processes should allow the large-scale production of bio-based products by cost-effective procedures to be carried out in sterile conditions, thus minimizing the risk of contamination [9]. Temperature is a critical parameter in industrial processes [10] that can influence the growth rate of microorganisms, the catalytic activity of enzymes, and the bioactivity of the molecules involved [11], thus affecting the overall performances of the process. High temperatures are widely used in different industrial sectors [12,13] for different purposes, such as food sterilization, but are also required in extraction procedures to improve extraction yields, as the heat improves the diffusion coefficient [14]; however, it must be taken into account that most of the molecules present in the microbial biomass are sensitive to high temperatures. This review aims to explore recent advancements in the use of thermophiles for developing biotechnological products addressing global challenges in sustainable production and environmental remediation. The focus is on harnessing thermophilic microbiomes, algae, and bacteria, particularly over the past five years, for applications in bioenergy, bioremediation, and the synthesis of valuable bioproducts (Figure 1).

## 2. Thermophilic Microbial Communities

Thermophilic microbiomes represent communities of microorganisms thriving in high-temperature environments, typically surpassing 45 °C [1]. These resilient microbes have acclimated to extreme heat and are commonly found in locales such as hot springs, geothermal vents, and compost heaps. Comprising a diverse array of life forms including bacteria, archaea, fungi, and other microscopic organisms, thermophilic microbiomes host notable taxa such as *Thermus*, *Sulfolobus*, *Thermococcus*, *Geobacillus*, and *Thermotoga*. Microbiomes have evolved specialized adaptations within these microbiomes to endure extreme thermal conditions [15]. These adaptations encompass the production of heat-stable enzymes, thermoprotective proteins, unique membrane lipids, and efficient DNA repair mechanisms [16]. As a result of these evolved thermal strategies, thermophilic microbiomes harbor immense potential for various applications in industrial biotechnology, offering promising avenues for research and innovation. Their heat-resistant enzymes and biochemical pathways hold promise for applications in bioenergy production, biocatalysis, bioremediation, and the degradation/synthesis of biopolymers [17,18,19]. In this section, we focus on harnessing thermophilic microbiomes from the past five years to develop biofuels and enhance bioplastic degradation to reduce pollution (Table 1).

### 2.1. Methane

In the pursuit of advancing the energy transition, biogas production through thermophilic anaerobic digestion emerges as a versatile renewable energy source, encompassing diverse feedstocks like biowaste and agricultural residues [20,21,22,23]. However, maintaining stability and optimizing biogas yields within the intricacies of the anaerobic digestion process presents significant challenges. Acknowledging the pivotal role of microbiomes within biogas digesters is crucial, as these microbial communities drive various stages of anaerobic digestion, from initial hydrolysis to methane production [24]. A notable study by Hassa et al. [25] employed a combination of metagenomics and metaproteomics to analyze the metabolic functional potential and the expressed functions in biogas digesters (Table 1). This study identified metagenomically assembled genomes (MAGs) adapted to varying digester conditions, resilient MAGs independent of digester conditions, and specific input materials along with their metabolic functions. The results underscored the importance of database deposition for better characterization of biogas microbiomes and understanding metabolic functions. However, limitations of MAG-centered metagenomics were noted, suggesting the need for deeper sequencing and additional omics technologies for a comprehensive understanding of biogas microbiomes [25]. Specifically, MAG-centered metagenome studies are limited to the more abundant microbiome members, as genomes of low-abundance or rare microorganisms are often not assembled and binned. This shortcoming can be addressed by deeper metagenome sequencing or the implementation of long-read sequencing technologies. Furthermore, incorporating other omics technologies, such as metatranscriptomics and metabolomics, would provide insights at all levels of the information flow from the genome sequence to metabolism. An integrative analysis of all omics data, combined with corresponding metadata, will allow for the reconstruction of metabolic networks, the identification of cooperating sub-microbiome assemblages, and an understanding of dependencies and interactions between biogas microbiome members [25]. Furthermore, integrating omics data with metadata is needed to enhance biogas digester monitoring and control strategies in the future. Another avenue for leveraging thermophilic microbiomes in biogas production involves utilizing beverage wastes, which are abundant in sugars and proteins. However, challenges such as over-acidification and methane production inhibition can arise from high substrate loading. Matsuda et al. [26] investigated changes in microbial communities during over-acidification in thermophilic methane fermentation using beverage waste, identifying key species involved in various stages of fermentation (Table 1). In particular, from the 16S rRNA amplicon sequence analysis, it was revealed that *Coprothermobacter proteolyticus*, *Defluviitoga tunisiensis*, *Acetomicrobium mobile*, and *Thermosediminibacter oceani* were predominantly involved in hydrolysis/acidogenesis/acetogenesis processes, whereas *Methanothrix soehngenii* was the major acetotrophic methane producer. Additionally, Lin et al. [27] compared the anaerobic hydrolysis and acidogenesis of pig manure at temperatures of 55 and 70 °C, revealing comparable performance between the two temperatures despite higher concentrations of certain compounds in the hyperthermophilic reactor. This suggests that thermophilic temperatures may offer energy consumption advantages over hyperthermophilic temperatures in industrial applications.

### 2.2. Medium-Chain Fatty Acids: n-Caproate

Recent studies have redirected the focus from methane towards medium-chain fatty acids (MCFAs) like *n*-caproate, utilizing a carboxylate platform, i.e., an undefined-mixed-culture process to generate a mixture of carboxylates as an intermediate of complex fuels [28]. MCFAs, containing 6 to 12 carbon atoms, hold greater value compared to traditional anaerobic products such as methane and short-chain fatty acids (SCFAs) [29]. MCFAs have diverse applications, serving as replacements for antibiotics in animal feed, flavoring agents in food, and antibacterial agents in pharmaceuticals [30,31]. As vital precursors for biofuels and chemical feedstocks, the refined carboxylate platform can achieve an economic value of USD 2000 to USD 3000 per ton [32]. In this context, Zhang et al. investigated the efficiency of converting food waste into *n*-caproate without the need for external electron donors, using a thermophilic microbiome in both batch and semi-continuous fermentation methods [32]. The efficiency of *n*-caproate production, its components, and microbial characteristics were examined at 55 ℃ to inhibit the growth of methane archaea. It was demonstrated that *n*-caproate production was strongly associated with lactate, which acted as an electron donor, facilitating chain elongation reactions. Analysis of the thermophilic microbial community identified *Caproiciproducens*, *Rummeliibacillus*, and *Clostridium_sunsu_stricto_12* as crucial genera involved in *n*-caproate synthesis, while *Clostridium_sensu_stricto_7* was primarily responsible for converting lactate into n-butyrate (Table 1) [32]. These insights contribute to the optimization of *n*-caproate production and enhance our understanding of the thermophilic microbial dynamics involved in the process, thereby paving the way for more sustainable waste utilization and value-added product synthesis.

### 2.3. Hydrogen

Thermophilic biohydrogen production holds promise as a sustainable solution for renewable energy generation, especially when coupled with industrial waste effluents rich in organic matter. Studies have shown that employing thermophilic microbial communities directly in hot effluents, such as those from food industries, can enhance hydrogen production rates. For instance, Pason et al. [33] demonstrated successful thermophilic H_2_ production using untreated cassava pulp as a substrate, achieving a maximum H_2_ yield of 760 mL/L at 60 °C. The genetic modification of thermophilic microorganisms and the application of nanotechnology are emerging as avenues to enhance H_2_ production. For instance, *Thermoacetogenium phaeum* and *Clostridium ultenense* are known for their ability to convert volatile fatty acids (VFAs) to H_2_ with external energy input. Genetic modifications aimed at optimizing metabolic pathways, such as deleting the lactate dehydrogenase (ldhA) gene and enhancing the alcohol dehydrogenase (adhA) gene, show potential for increasing H_2_ yield [34]. Additionally, the application of nanotechnology presents opportunities to develop sustainable catalysts for improving microbial growth and enzyme activity at higher temperatures. Studies have shown that nanomaterials, such as nickel ferrite nanoparticles, can enhance thermophilic H_2_ production yields by ∼28.3%, offering a pathway toward economically feasible biohydrogen production at commercial scales (Table 1) [35]. Overall, these studies underscore the importance of optimizing process parameters, substrate selection, and microbial communities to achieve efficient thermophilic biohydrogen production, paving the way for green and sustainable fuel solutions.

### 2.4. Biodegradable Plastics

The rise in environmental concerns surrounding traditional plastics has prompted the use of biodegradable plastics (BPs). However, questions remain about their biodegradability, particularly in anaerobic environments. In the study of Jin et al. [36], the anaerobic degradation of ten common BPs under mesophilic and thermophilic conditions was evaluated. Results showed the significant degradation of four BPs under mesophilic conditions (57.9–84.6% biodegradability) and notable degradation of five BPs under thermophilic conditions (53.0–95.7% biodegradability), through bulk or surface erosion mechanisms. Key microbial communities involved in degradation included *Anaerolineales*, *Bacteroidales*, *Clostridiales* SBR1031, and *Synergistales* under mesophilic conditions, while *Coprothermobacter* and *Methanothermobacter* are prominent under thermophilic conditions. Yu et al. [35] evaluated the degradation of three types of bioplastic bags during anaerobic co-digestion with food waste under mesophilic/thermophilic conditions. Results suggest that thermophilic co-digestion reduced lag time before methane production by one to four days and improved bioplastic conversion by 9.11–11.2% (Table 1). Analysis of the microbial community structure revealed that Archaea exhibited varying levels of biodiversity at different temperatures. At the genus level, the thermophilic group showed a diverse Archaea community with twelve genera having abundances greater than 1%, indicating significantly higher diversity. Specifically, under thermophilic conditions, the relative abundance of *Methanothermobacter* reached 56.0%, playing a crucial role in the anaerobic degradation of PBAT/PLA/starch materials. In contrast, bacterial communities showed smaller differences in abundance. Overall, the anaerobic co-digestion of bioplastic with food waste offers a renewable energy source. These studies offer insights into thermophilic microbial communities, which could be instrumental in developing customized biotechnological solutions for effective plastic degradation.

**Table 1 ijms-25-07685-t001:** Industrial applications of thermophilic microbiomes.

Collection Site	Location	Biomolecule	Temperature (°C)	Application	Reference
Biogas plant	Eastern Bavaria, Germany	methane	22–57	Biofuel	[25]
Waste beverage treatment plant	Shizuoka, Japan	methane	55	Biofuel	[26]
High-solids pig manure	Chinese Academy of Agricultural Sciences	methane	55 and 70	Biofuel	[27]
Recycling company	Jinan, China	*n*-caproate	55	Biofuel	[32]
Compost and soil sediments	n.r.	H_2_	60	Biofuel	[33]
Biogas station	Tongzhou, Beijing	methane	37 and 55	Bioplastic degradation	[35]

Abbreviation n.r. stands for not reported.

## 3. Thermophilic Bacteria

Thermophilic bacteria represent a powerful platform for biotechnological applications with their ability to thrive at high temperatures, thus reducing the risk of contamination and enhancing the enzymatic processes for bioconversion [37]. However, their unique metabolic capabilities permit them to efficiently convert raw materials into valuable bioproducts under conditions that are challenging for mesophilic organisms. Advancements in genetic engineering and bioreactor design are ready to further unlock the potential of thermophilic bacteria in industrial applications [38,39].

### 3.1. Hydrogen-Producing Bacteria

The production of hydrogen by thermophilic bacteria is a significant area of interest in renewable energy research. Thermophilic bacteria can convert organic substrates into hydrogen through various metabolic pathways, including dark fermentation. In this case, carbohydrates are degraded in the absence of light to produce hydrogen, carbon dioxide, and organic acids. The Embden–Meyerhof–Parnas pathway (EMP or glycolysis) is commonly used for the degradation of carbohydrates [40]. Specific species metabolize substrates into various types of sub-products, such as ethanol, acetate, lactate, CO_2_, and hydrogen, demonstrating the versatility of their metabolic pathways. For example, *Caldicellulosiruptor saccharolyticus* is known to degrade cellulose into hydrogen and other minor products such as acetate and carbon dioxide, almost reaching the theoretical limits of hydrogen production from sugars [40]. This bacterium is capable of producing hydrogen with yields ranging from 2.9 to 3.4 moles of hydrogen per mole of hexose used. These yields correspond to approximately 74–85% of the theoretical maximum, indicating a high efficiency in the production of hydrogen from lignocellulosic biomasses such as the flowering plant belonging to the grass family *Miscanthus sinensis*. *Thermotoga neapolitana* produces significant amounts of hydrogen through dark fermentation, thriving between 40 °C and 80 °C on various substrates, including simple and complex sugars and industrial residues [41]. *Clostridium thermocellum* shows a high efficiency in the conversion of cellulosic biomass into hydrogen. Research indicates that the use of membrane bioreactors for in situ gas extraction could significantly improve hydrogen production compared to conventional anaerobic fermentation setups, increasing both the rate and yield of hydrogen. Specifically, production increased from 25.8 to 42.1 mmol H_2_ using cellobiose as the carbon source and from 46.8 to 74.6 mmol H_2_ using Avicel (microcrystalline cellulose) [42]. These yields correspond to approximately 0.23 to 0.37 moles of hydrogen per mole of hexose for cellobiose and 0.42 to 0.67 moles of hydrogen per mole of hexose for Avicel (Table 2). Despite the promising results, challenges remain, such as the need for specialized bioreactor systems to handle high temperatures and optimize yields. Further research could include the genetic engineering of bacterial strains or the improvement in bioreactor design. Therefore, although thermophilic fermentation offers a promising and sustainable method of hydrogen production, further developments are needed to scale this technology to a commercial level. This requires a deep understanding of biological mechanisms, processes, and technological advances to develop more effective solutions for a sustainable energy future.

### 3.2. Biodegradable Plastics Production

In the field of sustainable materials, the employment of thermophilic bacteria in bioplastic production represents a significant breakthrough, particularly through their efficient transformation of renewable feedstocks into valuable polyhydroxyalkanoates (PHAs) [43,44]. Thermophiles leverage diverse carbon sources, such as lignocellulosic biomass, converting them into bioplastics with impressive efficiency. Research indicates that these bacteria can achieve PHA yields as high as 75% of their biomass under optimized conditions, with glucose frequently serving as the primary carbon source due to its robust conversion rate into PHB—a prevalent form of PHA [45]. Pioneering studies in thermophilic biotechnology have highlighted substantial advancements in the synthesis of PHAs, underscoring the role of thermophilic bacteria. For instance, strains such as *Cupriavidus cauae* PHS1 and *Geobacillus stearothermophilus* have demonstrated their potential to produce PHAs from various substrates at elevated temperatures, thereby minimizing contamination risks and enhancing yield [45]. *Cupriavidus cauae* PHS1, in particular, excels as an alternative to mesophilic PHA producers, effectively synthesizing polyhydroxybutyrate from substrates like acetate and phenol at 45 °C using gluconate. Furthermore, the *Geobacillus stearothermophilus* K4E3_SPR_NPP strain has been effectively utilized to generate PHA from dairy industry effluent, illustrating its capability to utilize low-cost substrates [45]. Similarly, *Aneurinibacillus* sp. H1 adeptly transforms glycerol into poly(3-hydroxybutyrate) and copolymers containing significant molar fractions of 3-hydroxyvalerate and 4-hydroxybutyrate under optimal conditions at 45 °C [46,47]. Meanwhile, *Schlegelella thermodepolymerans* DSM 15344, a moderately thermophilic bacterium, reveals a genome indicating metabolic versatility that allows it to process a variety of substrates such as lignocelluloses and glycerol for PHA production [48]. The use of lactic acid as a precursor further exemplifies the potential for sustainable bioplastic production. Thermophilic bacteria such as *Weizmannia coagulans* MA-13 are recognized for efficiently converting sugars from lignocellulose into lactic acid [49,50]. This lactic acid can subsequently be polymerized into polylactic acid (PLA), highlighting an innovative pathway for the sustainable production of bioplastics using non-food raw materials [50,51,52,53]. Another noteworthy bacterium, *Caldicellulosiruptor* sp. strain DIB 104C, initially isolated for its ethanol production capabilities from lignocellulose, also exhibits significant potential for lactic acid production from microcrystalline cellulose and lignocellulosic biomass. This strain can convert a variety of sugars into lactic acid, with minimal byproduct formation, making it a promising candidate for a streamlined lactic acid production process [53] (Table 2). These findings collectively illustrate the vast potential of leveraging thermophilic bacteria and renewable resources in the creation of environmentally friendly bioplastics. This approach addresses agricultural waste management and presents a sustainable alternative to traditional petroleum-based plastics, marking a pivotal development in sustainable manufacturing and offering a promising solution to the global challenge of plastic pollution.

### 3.3. Plastic Biodegradation

The relentless accumulation of plastic waste in the environment has prompted the urgent need for innovative and sustainable bioremediation strategies. Recent advancements have highlighted the potential of thermophilic microbes in addressing this challenge. In a significant advancement for plastic bioremediation, a thermophilic strain of *Clostridium thermocellum* expresses a cutinase enzyme, known as leaf-branch compost cutinase (LCC), capable of degrading polyethylene terephthalate (PET) at elevated temperatures. This thermophilic whole-cell biocatalyst demonstrated a marked improvement in PET degradation, achieving over 60% mass conversion of commercial PET films into soluble monomers within 14 days at 60 °C. The study’s findings suggest that thermophilic microbes, owing to their optimal growth and enzyme activity at higher temperatures, present a more promising PET biodegradation approach than mesophilic organisms, offering significant potential for advancing sustainable plastic waste management strategies. Similarly, Skariyachan et al. [54] explore an innovative and sustainable method for the biodegradation of low- and high-density polyethylene (LDPE and HDPE) using thermophilic bacterial consortia isolated from plastic-contaminated cow dung. These bacteria, identified as *Bacillus vallismortis* bt-dsce01, *Pseudomonas protegens* bt-dsce02, *Stenotrophomonas* sp. bt-dsce03, and *Paenibacillus* sp. bt-dsce04, demonstrated remarkable degradation capabilities, achieving 75% degradation for LDPE strips and 60% for HDPE strips in 120 days at 55 °C [54]. Analysis of the end products using FTIR, SEM, EDS, and NMR revealed significant structural modifications in the polymers and the formation of bacterial biofilms. This approach promises to revolutionize plastic waste management, offering an ecological and potentially scalable solution for the rapid removal of polyethylene from the environment.

Another study on the biodegradation of plastics explored the degradation of poly(ε-caprolactone) (PCL) using a thermophilic microbial community from the Marikostinovo hot spring in Bulgaria and a strain of *Brevibacillus thermoruber* [55]. The research identified a strong dominance of the phyla Deinococcus-Thermus and Firmicutes in the PCL-enriched microbial community, with *Brevibacillus thermoruber* strain 7 showing the highest esterase activity at 55 °C. The biodegradation process, observed over 28 days, resulted in significant reductions in molecular weight and weight loss of PCL, achieving complete degradation by the microbial community and 63.6% by strain 7 (Table 2). SEM analysis revealed extensive surface deformation and biofilm formation by the microbial community, whereas the pure strain caused significant plastic deformation without biofilm formation. This study highlights the potential of thermophilic microbes in plastic waste management, offering insights into the microbial and enzymatic processes involved in PCL degradation. These studies collectively underscore the promising role of thermophilic microbes in the biodegradation of various types of plastics, including PET, LDPE, HDPE, and PCL. By leveraging the unique metabolic pathways and enhanced enzyme activities of thermophiles, efficient and sustainable methods for mitigating plastic pollution can be developed. The continued exploration and optimization of these microbial consortia and engineered strains hold significant potential for advancing global plastic waste management efforts and reducing the environmental footprint of plastic materials.

**Table 2 ijms-25-07685-t002:** List of thermophilic bacteria employed in industrial applications.

Strain	Collection Site	Biomolecule	Thermal Stability (°C)	Application	Reference
*Caldicellulosiruptor saccharolyticus*	Thermal spring, New Zealand	H_2_	70–75	Hydrogen production	[40]
*Thermotoga neapolitana*	Thermal spring, Italy	H_2_	50–95	Hydrogen production	[41]
*Clostridium thermocellum*	Agriculture residues and wastes, Thailand	H_2_	60	Hydrogen production	[42]
*Cupriavidus cauae* PHS1	Thermal spring, Korea	PHAs	45	Biodegradable plastics production	[43]
*Geobacillus stearothermophilus* K4E3_SPR_NPP	Kasol Hot Spring, India	PHAs	70	Biodegradable plastics production	[45]
*Aneurinibacillus* sp. H1	Compost, Czech Republic	PHAs	45	Biodegradable plastics production	[46]
*Schlegelella thermodepolymerans* DSM 15344	n.r.	PHAs	55	Biodegradable plastics production	[48]
*Weizmannia coagulans* MA-13	Bean waste	Lactic acid	55	Biodegradable plastics production	[49,50,51,52]
*Caldicellulosiruptor* sp. strain DIB 104C	n.r.	Lactic acid	55	Biodegradable plastics production	[53]
*Clostridium thermocellum*	n.r.	n.r.	60	PET degradation	[42]
*Bacillus vallismortis* bt-dsce01	Plastic-contaminated environments, India	n.r.	55	LDPE and HDPE degradation	[54]
*Pseudomonas protegens* bt-dsce02	Plastic-contaminated environments, India	n.r.	55	LDPE and HDPE degradation	[54]
*Stenotrophomonas* sp. bt-dsce03	Plastic-contaminated environments, India	n.r.	55	LDPE and HDPE degradation	[54]
*Paenibacillus* sp. bt-dsce04	Plastic-contaminated environments, India	n.r.	55	LDPE and HDPE degradation	[54]
*Brevibacillus thermoruber* strain 7	Hot spring, Bulgaria	n.r.	55	PCL degradation	[55]

Abbreviation n.r. stands for not reported.

## 4. Thermophilic Microalgae and Cyanobacteria

Microalgae and cyanobacteria are a group of diverse photosynthetic microorganisms able to fix CO_2_ to produce chemical energy and O_2_ [56,57]. CO_2_ sequestration plays a key role in reducing greenhouse gases, and thus, microalgae can be considered a sustainable tool to preserve the environment. Along with CO_2_ capture, microalgae and cyanobacteria can synthesize a wide array of bioactive molecules, such as lipids, proteins, carbohydrates, and carotenoids during photosynthesis [58,59]. They are considered a reliable and continuous source of high-value compounds for these reasons. Due to the presence of these molecules in algal biomass, in recent years, microalgae and cyanobacteria have been studied for biotechnological applications. Based on their nature, lipids can be used to produce renewable fuels and food supplements (polyunsaturated fatty acids), whereas other bioactive molecules are endowed with anticancer, antioxidant, and anti-inflammatory activities [60,61]. In recent years, thermophilic microalgae and cyanobacteria gained attention for their biotechnological and industrial potential not only for thermostable enzymes but also for high-value molecules (i.e., phycobiliproteins and carotenoids), biopolymers, biofuels, wastewater treatment, and for bioremediation [62,63]. An overview of the microalgae and cyanobacteria and their applications discussed in this review is reported in Table 3.

### 4.1. Thermophilic Phycocyanin

Phycocyanin (PC), along with Phycoerythrin (PE) and allophycocyanin (APC), is a protein that belongs to phycobiliproteins (PBPs), which are colored and naturally fluorescent proteins that can be mostly found in Cyanobacteria, Rhodophyta, and Cryptomonads [72]. PBPs are assembled in a multimeric structure, the phycobilisome, that serves as a light-harvesting antenna complex, thus improving the photosynthetic efficiency [73]. PE and PC have been extensively studied as they can exert a wide range of biological activities. It has been reported that PBPs are endowed with antioxidant, anticancer, anti-inflammatory, and anti-neurodegenerative activity [74,75]. Indeed, they can find different applications both in the pharmaceutical and cosmetic industries [76]. Due to their natural color and high fluorescence, PBPs can be used as fluorescent probes in research laboratories and as natural dyes in the food industry. Moreover, due to their biological activities, PBPs can also be used as food additives. However, the increase in temperature normally required in pasteurization or in other sterilization processes results in color deterioration [77] and a loss of biological activities, as proteins undergo denaturation processes, thus undermining the use of these proteins at an industrial level [78]. To date, the most commercialized PC is isolated from *Spirulina* [79]. To increase its thermal stability, several stabilizers have been proposed, such as chemical preservatives, sugars, antioxidant molecules, or sugars and salts; however, some of these compounds were found to be toxic at the optimal stabilizing concentrations [78]. In this scenario, thermophilic microalgae and cyanobacteria have a substantial advantage over mesophilic strains in terms of their ability to biosynthesize robust phycobiliproteins [62]. Cyanidiales, a group of red microalgae that are classified into *Cyanidium*, *Cyanidioschyzon*, and *Galdieria* stains, have emerged as the most promising strains for the obtainment of highly stable PC both at lab and industrial scales [80,81].

Ferraro et al. [8] deeply investigated the physicochemical properties of a highly thermostable PC isolated from *Galdieria phlegrea*. This study revealed that the purified PC had a melting temperature (Tm) of 87 °C, at pH 5.5 and 7.0, and the protein was able to retain the tertiary structure at 77.5 °C, being stable for more than 1 h at 75 °C. Moreover, PC was able to retain antioxidant activity upon pasteurization. By resolving the structure, they found out that this enhanced thermostability was due to a significantly higher number of ionic interactions at the (αβ)_3_/(αβ)_3_ interface, as a result of mutations of amino acid residues [82]. Yoshida et al. [64] examined the thermostability of the PC extracted from *Cyanidioschyzon merolae* at different salt solutions, proving that the protein maintained its thermal stability up to 60 °C, with a Tm of 83 °C. Rahman and colleagues [65] obtained PC from a *Galdieria sulphuraria* 074G growth on a maltodextrin and granular starch-enriched medium to improve the productivity and extraction yield. PC exposed to different temperatures (30–80 °C) for 30 min was stable up to 55 °C, as no changes in color and precipitation events were observed; however, at 60 °C, most of the protein precipitated and a loss of color was observed. Overall, the obtainment of stable PCs can face several challenges at an industrial level, such as the retention of the color and the bioactivity and the extension of the shelf-life of the products, and it can represent an advantage for long-term storage. For these reasons, understanding and improving the thermostability of phycocyanin is fundamental as it directly influences its practical applications and marketability.

### 4.2. Lipids

The extensive use of fossil fuels is one of the principal causes of global warming and the depletion of non-renewable energy sources, thus raising serious concerns and the urgency to push the transition toward a more sustainable future. Over years, biofuels, such as biodiesel, bioethanol, biohydrogen, biobutanol, and biogas, have emerged as a green alternative to conventional fuels [83]. Biodiesels are fatty acid alkyl esters that can be obtained from natural and renewable sources [84]. In this scenario, microalgae and cyanobacteria attracted interest as they can accumulate up to 50% lipids within their biomass [85]. Microalgae and cyanobacteria possess a photosynthetic efficiency and biomass productivity superior to traditional oil crops; they can be cultivated in non-arable lands, using wastewater, thus reducing the competition with food production and the impact on the environment; they are able to remove CO_2_ from the environment; and they can be cultivated in both open and closed systems, indoors and outdoors, thus overcoming the problem of seasonality [57,86]. The advantage of using thermophilic microalgae in biodiesel production is the possibility to cultivate them in outdoor systems, avoiding contaminations. Another advantage relies on the ability of these microorganisms to enhance lipid biosynthesis when cultivated at an elevated temperature [87,88]. Singh et al. [66] carried out an interesting screening on 52 cyanobacteria and 57 microalgae strains isolated from both freshwater and hot spring water in the region of Garhwal Himalaya (Uttarakhand, India) to be used as biodiesel feedstock. This study proved that, among all the strains under test, *Pseudobohilina* sp. (PbS–BHS) and *Letpolynbgya foveolarum* (LlF-RHS), two strains isolated from hot springs, and thus able to tolerate high temperature, were found to be the most promising strains in terms of biomass productivity and lipid accumulation. The same author investigated the effect of the temperature on another strain of *Leptolyngbya*, *L. foveolarum* HNBGU001, a cyanobacteria able to grow prevalently in subtropical and tropical environments. The optimal growth conditions were found to be 40 °C and pH 8.0; indeed, *L. foveolarum* HNBGU-001 showed a biomass productivity, a lipid content, and a lipid productivity 2.8-, 2.4-, and 6.8-fold higher, respectively, compared to the non-optimized growth conditions. Finally, fatty acid methyl ester (FAME) analysis confirmed promising features for biodiesel production [67]. Boutarfa et al. [68] studied three different strains of *Mastigocladus laminosus* Cohn ex Kichner, a cyanobacterium isolated from hot springs in Algeria, a potential biodiesel feedstock. In particular, the lipid profiles were defined and different biodiesel properties, such as the degree of unsaturation, oxidative stability, iodine value, density, saponification value, kinematic viscosity, and others, were evaluated. Lipid profiles had a ratio between saturated fatty acids (SFAs) and monounsaturated fatty acids (MUFAs) of 55.91–59.37%, with an average chain length of C14 to C20. Overall, the results of this study revealed the suitability of *M. laminosus* for biodiesel production as all the parameters under test fulfill the international standards.

### 4.3. Exopolysaccharides

Exopolysaccharides (EPSs), along with other molecules, are involved in the adaptation process of thermophiles. EPSs are high-molecular-weight polymers composed of repeated and arranged sugar residues that are generally secreted into the environment and that surround cells. In extreme conditions, EPSs increase the adhesion between cells, by preventing dehydration [89]. EPSs from microalgae and cyanobacteria have gained attention for their structural features, such as the presence of sulfate esters attached to glycosides [90] that confer to them unique biological activity, such as antioxidant, anti-inflammatory, and antiviral properties [59]. Moreover, they can be used as emulsifiers, thickening, and stabilizer agents [91]. EPSs can also serve as a sustainable and economic alternative to conventional flocculants, and due to their ability to bind heavy metals, EPSs can be used for bioremediation. EPSs from extreme environments offer a diversity of physicochemical properties, such as improved thermal stability [92], a plethora of biotechnological features, such as short fermentation processes [89], and non-pathogenicity, thus making these polymers suitable for the development of food, cosmetic, and pharmaceutical products [93]. Moreover, temperature is a key parameter that has a strong influence on microalgae metabolism; indeed, high temperature can enhance EPS production [94]. Recently, Gudmundsdottir and colleagues investigated the anti-inflammatory role of EPSs produced by *Cyanobacterium aponinum* [69], a thermophilic cyanobacterium grown at 40 °C [95]. The research was mainly devoted to elucidating the molecular mechanism that underlies psoriasis, using dendritic cells, keratinocytes, and T cells as experimental systems. In detail, EPS treatment was able to induce a regulatory phenotype and decrease the activity of T cells. In keratinocytes, EPSs were able to reduce the production of chemokines and one of the autoantigens involved in psoriasis. Gongi isolated and investigated EPSs from the thermophilic cyanobacterium *Gloeocapsa gelatinosa*, defining the physicochemical properties and the biological activity [70]. In particular, this study elucidates that EPSs were composed of two anionic polymer fractions with different molecular weights, and they were found to be thermostable at temperatures higher than 100 °C. Among other characteristics, EPSs from *G. gelatinosa* were endowed with higher water holding capacity and a high solubility index, properties fundamental for the food industry as they are related to a possible use as a fat adsorber and for flavor retention, and a strong Cu^2+^ and Fe^2+^ sorption capacity, thus conferring them a flocculant ability to be used in wastewater treatment. Finally, the in vitro analyses revealed good free radical scavenging properties, so that EPSs from this cyanobacterium may also serve as novel natural antioxidants. The same research group performed a similar investigation on EPSs isolated from another thermophilic cyanobacterium from a Tunisian hot spring, *Leptolyngbya* sp. IkmLPT16 [71]. EPSs from this strain were sulfated heteropolysaccharides that showed good functional properties when analyzed, such as water and oil holding capacity, and the ability to absorb metal ions. The rheological properties resembled the ones of the commercial xanthan gum with higher resistance; however, EPSs from *Leptolyngbya* sp. showed a higher thermostability (up to 60 °C). Finally, the antioxidant activity was demonstrated by in vitro assays. The unique properties, sustainable production, and stability at high temperatures make EPSs from microalgae and cyanobacteria a valuable resource for driving innovation and sustainability across numerous industrial fields.

## 5. Conclusions

The study of thermophiles across various life domains sheds light on their exceptional ability to thrive in extreme temperatures, providing valuable insights into biodiversity, evolutionary history, and adaptation mechanisms. Over the past five years, thermophilic microbiomes and bacteria, along with thermophilic microalgae and cyanobacteria, have emerged as promising assets for industrial biotechnology, renewable energy, waste management, and environmental sustainability. Although their unique features could drive innovation across several industrial fields, still several challenges must be addressed, such as the optimization of bioprocesses, the exploration of novel thermophilic species, and the integration of biotechnological approaches with synthetic biology and metagenomics, so that innovative biotechnological solutions can be unlocked to pave the way for innovative solutions to global challenges.

## Figures and Tables

**Figure 1 ijms-25-07685-f001:**
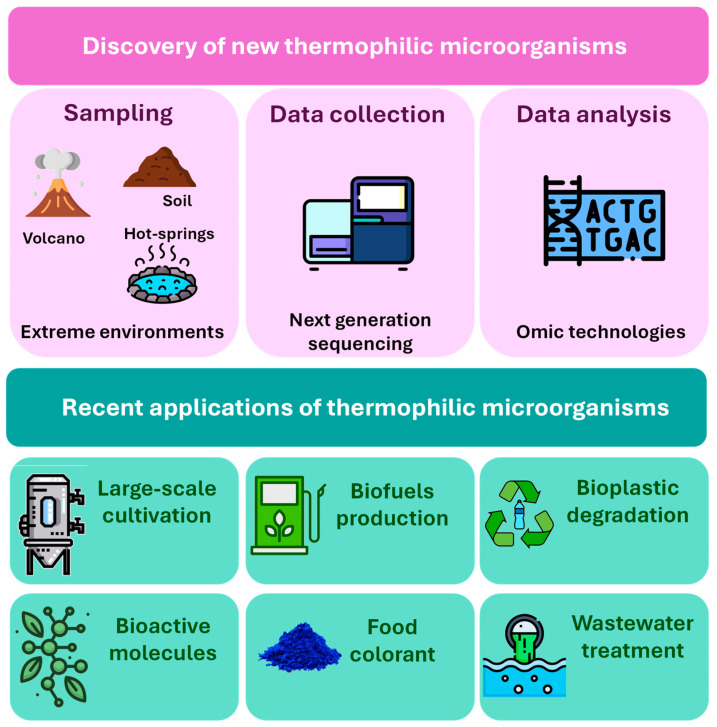
Exploration and utilization of thermophilic microorganisms for industrial applications.

**Table 3 ijms-25-07685-t003:** Application and thermal stability of molecules extracted from thermophilic microalgae and cyanobacteria.

Strain	Collection Site	Biomolecule	Thermal Stability (°C)	Application	Reference
*Galdieria phlegrea*	Sulphur spring, Italy	PC	87	Antioxidant, anticancer, natural dye/additive	[8]
*Cyanidioschyzon merolae*	n.r.	PC	83	Natural dye	[64]
*Galdieria sulphuraria* 074G	n.r.	PC	55	Large-scale cultivation	[65]
*Pseudobohilina* sp. PbS–BHS	Badrinath hot spring, Chamoli, Garhwal	Lipids	n.r.	Biodiesel	[66]
*Leptolyngbya foveolarum* LlF-RHS	Ringigad hot spring, Chamoli, Garhwal	Lipids	n.r.	Biodiesel	[66]
*Leptolyngbya foveolarum* HNBGU001	Thermal springs, Garhwal, Himalaya	Lipids	n.r.	Biodiesel	[67]
*Mastigocladus laminosus* Cohn ex Kichner	Hot spring, Algeria	Lipids	n.r.	Biodiesel	[68]
*Cyanobacterium aponinum*	Geothermal pool, Blue Lagoon, Iceland	EPSs	n.r.	Anti-inflammatory	[69]
*Gloeocapsa gelatinosa*	Hot source, Ain Echfa, Tunisia	EPSs	100	Antioxidant; fat adsorber; flavor retention; wastewater treatment	[70]
*Leptolyngbya* sp. IkmLPT16	Hot water, Ain Echfa, Tunisia	EPSs	60	Antioxidant	[71]

Abbreviation n.r. stands for not reported.

## Data Availability

Not applicable.

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
