# Peer review of "The Undeniable Potential of Thermophiles in Industrial Processes"

_ijms, 2024, doi:10.3390/ijms25147685_

Round 1

Reviewer 1 Report

Comments and Suggestions for Authors

Overall, I had a very favorable impression of this review article.  It covered a broad range of applications and was informative. I have a couple of general comments for improving the overall article. Given the widespread use and potential of thermophiles  in bioremediation and biomining of heavy metals, I wonder if a section should be included here?

In places the writing makes the assumption that the reader knows about these organisms and about the processes in question. Because this is a review article, care should be taken to make sure that the points are accessible to a general audience rather than just those in the field. The text would also benefit from careful editing for grammar and clarity. I have pointed out a few specific examples below.

Line 172 replace “of” with “by”.

 Line 189  “moreover” is not needed.

Line 193-195  The connection between Methanothermobacter and the bacterial community is unclear as presented.

Line 201  In the table title replace “List” with “Uses” or “Applications”

Line 205  The distinction between this heading and the 2.0 is unclear.

Line 376-368. Starting the sentence with “Mostly” is awkward in this situation.

Line 373.  It should read: When proteins are exposed to heat,…

Line 373-375. The comment about sterilization/pasteurization needs to be explain. Not all readers will be able to follow this. For example, is this a desirable or undesirable property?

Line 389. Could be changed to: PC was able to retain antioxidant activity upon pasteurization.

Line 434. Should read: had a ratio between saturated…

Line 443-445. Awkward sentence

Line 460. Singular-plural mismatch

Line 483. I believe this sentence should be removed.

Comments on the Quality of English Language

The manuscript needs another round of editing.

Reviewer 2 Report

Comments and Suggestions for Authors

The authors expose a solid and interesting review about the role of thermophiles in biotechnology. The review touches the main applications of the thermophilic microorganisms, comparing with mesophilic microorganisms, and highlighting the added value of the achieved processes or compounds. In the next lines I share some comments and suggestions that, in my opinion, could improve the manuscript.

Major comments

1.    Lines 116-130. The authors could improve the readability of this paragraph, including more detailed explanations of some parts. For instance, when authors say “However, limitations of MAG-centered metagenomics were noted…”, I need to go to reference 25 to know about the limitations. It would be desirable to know the mentioned limitations in the text, in a couple of lines. In the same line, when authors say “However, challenges such as over-acidification and methane production inhibition…”, it would be desirable to know the main responsible in this over-acidification and inhibition of methane production.

2.    Line 134. The authors mention “the carboxylate platform” and it would be necessary to explain briefly what is that, considering that not everybody will be familiarized with this term.

3.    Line 155. In this section about hydrogen, it is not totally clear the benefit achieved by the use of thermophiles in the production of H2. On the one hand, the H2 yield of 760 mL/L is not compared with the yield obtained with mesophilic bacteria. Is it better, is worst, is the same? But maybe, the main point is that the effluent where these bacteria are used is so warm that is not possible to use mesophilic bacteria. In any case, is not clear the benefits compared with mesophilic bacteria.

4.    Lines 227-241. The authors talk about several bacteria species with the ability to produce H2. However, they give the yields just in some cases: 2.9-3.4 moles of H2 per mole of hexose in Caldicellulosiruptor saccharolyticus, “significant amounts of hydrogen” in the case of Thermotoga neapolitana, 25.8 to 42.1 mmol H2 using cellobiose as the carbon source and 46.8 to 74.6 mmol H2 using Avicel in the case of Clostridium thermocellum. Please, add the data of yields in all cases and use the same units to check easily the abilities of the different bacteria, considering that this is one of the roles of a review, to access the inform easily and rapidly.

5.    Lines 399 and 481. I think that these paragraphs talking about PC and EPSs, respectively, would need a final conclusion or final remark about the thermostable PCs and EPS, respectively.

Minor comments

-          Line 20: change “application” by “applications”.

-          Table 1: change “Agricoltural” by “Agricultural”. Write in italics the “n” of “n-caproate”, and do the same in all “n-caproate” or other similar molecules along the text.

-          Lines 106 and 108: remove the words “thorough” and “through”, respectively, to improve the comprehension.

-          Lines 227-229: the authors say “…from lignocellulosic biomasses such as Miscanthus sinensis”. Please, clarify this sentence to understand that this is a plant. For instance, “…from lignocellulosic biomasses such as the flowering plant belonging to the grass family Miscanthus sinensis”.

-          Please, remove the italics from “sp.” in the name of species that include it. Moreover, add italics in the species that need it, such as in line 296.

-          Lines 295-298: this sentence requires a reference.

-          Table 3: change “Island” by “Iceland”.  

-          Line 376: add a reference after “Spirulina”.

-          Lines 385, 452 ,465: change “physiochemical” by “physicochemical”.

-          Lines 387-388: the sentence is confused. I propose the next: “…and the protein was able to retain the tertiary structure at 77.5 °C, being stable for more than 1 h at 75 °C”.

-          Line 392: change “aminoacid” by “amino acid”.

-          Line 466: change “elucidate” by “elucidates”.

Comments on the Quality of English Language

The review requires a moderate English editing. I have proposed some changes in the previous comments but it would be recommendable to performance a professional editing.
